# Epigenetic Regulation of Auxin Homeostasis

**DOI:** 10.3390/biom9100623

**Published:** 2019-10-18

**Authors:** Eduardo Mateo-Bonmatí, Rubén Casanova-Sáez, Karin Ljung

**Affiliations:** Department of Forest Genetics and Plant Physiology, Umeå Plant Science Centre, Swedish University of Agricultural Sciences, SE-901 83 Umeå, Sweden

**Keywords:** epigenetics, auxin biosynthesis, auxin transport, auxin homeostasis

## Abstract

Epigenetic regulation involves a myriad of mechanisms that regulate the expression of loci without altering the DNA sequence. These different mechanisms primarily result in modifications of the chromatin topology or DNA chemical structure that can be heritable or transient as a dynamic response to environmental cues. The phytohormone auxin plays an important role in almost every aspect of plant life via gradient formation. Auxin maxima/minima result from a complex balance of metabolism, transport, and signaling. Although epigenetic regulation of gene expression during development has been known for decades, the specific mechanisms behind the spatiotemporal dynamics of auxin levels in plants are only just being elucidated. In this review, we gather current knowledge on the epigenetic mechanisms regulating the expression of genes for indole-3-acetic acid (IAA) metabolism and transport in *Arabidopsis* and discuss future perspectives of this emerging field.

## 1. Introduction

Phytohormones are small molecules naturally present in a very low abundance which orchestrate plant decisions during their entire life. Among them, indole-3-acetic acid (IAA), the main auxin form, is by far the most studied phytohormone. Many different developmental aspects, including tropisms, flowering time, circadian clock, and responses to biotic and abiotic stresses, have been linked to IAA [1,2,3,4]. Over the last few decades, extensive research has yielded a good understanding of how and where IAA is synthesized and how its polar transport generates local maxima/minima that trigger morphogenetic programs. Auxin homeostasis builds upon an intricate balance of biosynthesis, transport, conjugation, and degradation. Currently, a substantial number of genes are known to be involved in these processes, and in many cases, these genes are interconnected through feedback loops with other auxin-related genes and/or with components from other phytohormone pathways [5,6,7,8].

Auxin is perceived in the nucleus, where it negatively regulates inhibition of AUXIN RESPONSE FACTORs (ARFs), a group of transcription factors which modulate auxin-responsive genes. When the auxin concentration is low, ARFs associate in a repressive complex with Aux/IAA proteins and TOPLESS (TPL), which prevents ARF activity. Upon an auxin increment, IAA works as a molecular glue between TRANSPORT INHIBITOR RESPONSE 1 (TIR1), a member of the ubiquitin-protein ligase complex SCF^TIR1^, and Aux/IAAs proteins, tagging the latter for degradation and allowing ARFs to bind to gene regulatory regions containing auxin response elements (AuxRE) [9].

Ultimately, the transcriptional state of these genes is modulated through epigenetic processes that alter the accessibility of the transcriptional machinery to their regulatory regions. In *Arabidopsis*, over 130 genes encode components related to the epigenetic machinery, which can be grouped into five categories: (1) enzymes catalyzing chemical modifications of DNA or (2) histones, (3) Polycomb Group (PcG) and Polycomb-related proteins, (4) chromatin remodelers, and (5) protein and RNA molecules that participate in RNA-directed DNA methylation (RdDM) [9,10,11]. The current model of how auxin affects chromatin dynamics indicates that when auxin is not sensed, the whole complex (ARF-Aux/IAA-TPL) binds to AuxRE, recruiting a histone deacetylase complex (HDAC) and thus promoting chromatin compactness [12]. Conversely, the presence of auxin triggers Aux-IAA degradation, allowing ARFs to recruit histone acetyltransferases to promote chromatin opening in the target gene [12]. Recent findings support and complement this model: upon auxin stimuli, ARF5 recruits the chromatin-remodeling ATPases BRAHMA (BRM) and SPLAYED to genes involved in flower primordium initiation, such as *FILAMENTOUS FLOWERING*, *TARGET OF MONOPTEROS 3,* and *AINTEGUMENTA*. This chromatin unlocking allows recruitment of additional transcription factors and histone acetyltransferases [13]. However, ARFs do not always promote gene expression and, therefore, different epigenetic mechanisms are required depending on the ARF. Indeed, ARF3 and ARF4 have been found to repress *SHOOTMERISTEMLESS* (*STM*) via histone deacetylation, which promotes flower initiation at the reproductive meristem [14].

The relatively good understanding of chromatin dynamics associated with the auxin signaling machinery contrasts with a general lack of knowledge about the epigenetic regulation of genes controlling auxin homeostasis. Indirect evidence for the role of histone marks (e.g., H3K27me3) was first obtained in genome-wide comparative studies. These datasets suggested that a deep transcriptional reprogramming of genes for auxin biosynthesis and transport occurs during cell fate transitions [15,16].

In recent years, members of the five categories of chromatin rearrangement have been found to participate in regulating auxin metabolism- and transport-related genes. In this review, we summarize recent findings related to the epigenetic regulation of these genes and describe how this regulation drives different developmental programs. We also discuss future perspectives in this exciting and emerging field.

## 2. Epigenetic Control of Auxin Biosynthesis and Inactivation

Many reports have described how auxin polar transport is essential for generating local maxima and minima that trigger its morphogenetic effects. However, recent literature has highlighted the importance of local auxin biosynthesis and catabolism for root meristem maintenance [17]; root hair development [18]; embryo patterning [19]; pollen development [20]; as well as responses to shade [21,22], ethylene [17], and aluminum stress [23,24], among others. 

Auxin biosynthesis has been studied for decades, and although there are some gaps in our knowledge of the different pathways, the major routes are well understood (see reviews [1,2]). The most abundant auxin, IAA, is synthesized from the amino acid tryptophan (Trp). Three different intermediate precursors of auxin, namely, indole acetaldoxime (IAOx), indole acetamide (IAM), and indole-pyruvic acid (IPyA), are generated from Trp. IAOx is also a precursor of defense compounds, such as indole glucosinolates and camalexin, linking defense responses to IAA synthesis. The IAM pathway is well characterized in bacteria, but so far, no orthologues catalyzing the Trp conversion to IAM have been found in plant genomes. The best characterized route is the two-step IPyA pathway, in which members from two redundant enzyme families catalyze the conversion of Trp to IAA. TRYPTOPHAN AMINOTRANSFERASE OF ARABIDOPSIS1 (TAA1), together with TAA1-RELATED proteins (TARs), catalyzes the conversion of Trp to IPyA, whereas the flavin-containing monooxygenases YUCCA (YUC) transform IPyA into IAA [1,2].

Local and time-specific auxin requirements depend upon dynamic “on” and “off” switches of the biosynthetic machinery, driven by epigenetic processes. The first shreds of evidence for this regulation were found in comparative genome-wide occupancy studies of the histone repressive mark H3K27me3 [15,16]. These experiments revealed a differential pattern of this variant in genes involved in auxin biosynthesis, transport, and signaling between meristematic and differentiated cells [15] and between leaf and leaf-derived callus [16], suggesting global epigenetic control of auxin action. Specifically, *YUC1* and *YUC4* were found to be involved in early auxin-mediated de novo root regeneration, and this was correlated to a decline in H3K27me3 levels around their promoter regions, indicative of de-repression [25]. Furthermore, a member of the Polycomb Repressive Complex 2 (PRC2), LIKE HETEROCHROMATIN 1 (LHP1), was found to directly bind to a subset of *YUC* gene promoters, thus controlling their expression [26]. During the past five years, several lines of evidence have demonstrated a key role of epigenetic regulation in auxin homeostasis during different developmental contexts. Remarkably, most of the works have focused on this regulation during reproductive processes, such as floral determinacy, floral patterning, and seed development, but cases related to leaf development and responses to abiotic stress have also been reported.

Floral determinacy is the process by which cells from floral meristems stop proliferating and initiate a floral organ primordium. Regulation of this balance is essential for plant reproductive success and requires the integration of endogenous and exogenous cues. The master regulator of flower determinacy is *AGAMOUS* (*AG*), a MADS-box transcription factor that acts by direct targeting of other transcription factors, such as *WUSCHEL*, *KNUCKLES*, and *CRABS CLAW* (*CRC*) [27,28,29,30]. The first steps of flower primordia formation require rapid cell expansion and elongation correlating with an increment of auxin concentration. Recently, it was shown that both CRC and AG activate *YUC4* expression in flower primordia [31]. Expression analyses revealed that *ag* and *crc* mutants have lower levels of *YUC4* mRNA in flower primordia, as well as lower amounts of IAA. It was also demonstrated that both CRC and AG bind to the *YUC4* promoter through the YABBY binding site and CArG box, respectively. Using the formaldehyde-assisted isolation of regulatory elements (FAIRE) assay, it was shown that AG recruits the chromatin-remodeling factors CHROMATIN REMODELING 11 (CHR11) and CHR17 to the *YUC4* promoter. This recruitment allows CHR11 and CHR17 to open the *YUC4* chromatin, which promotes subsequent access of the RNA polymerase II (Figure 1a, Table 1) [31].

A complex interplay between members of several families of transcription factors, as well as the action of different hormones such as auxin and brassinosteroids, occurs at the floral meristem–organ boundaries to define flower patterning [32]. This pattern results from coordinated responses in cells from proliferating flower meristems, flower primordia, and meristem–primordia boundaries [33,34]. One of the genes involved in this process is *SUPERMAN* (*SUP*), which encodes a transcription factor with a C2H2-type zinc finger motif expressed at the boundaries between stamen primordia and carpels. Lately, it has been shown that mutations in *SUP* lead to higher levels of auxin in these boundaries, resulting in flowers with supernumerary stamens. The SUP protein actively represses auxin biosynthesis at the boundaries by physically associating to *YUC1* and *YUC4* promoters and recruiting the PRC2 machinery to mediate the trimethylation of H3K27. Based on protein–protein interaction experiments, it has been demonstrated that this recruitment is carried out by direct interaction of SUP with CURLY LEAF (CLF) and LHP1 (Figure 1b, Table 1) [35].

The final purpose of all these complex regulatory networks during flower development is to ensure development of viable seeds to transmit the genetic information to the next generation. Angiosperm seed development requires a singular event of double fertilization, in which a pollen grain (male gametophyte) releases two sperm cells—one fuses to the female egg, whereas the other fuses to the diploid central cell. The fertilized egg subsequently forms the plant embryo, which grows within the endosperm, a triploid tissue that results from the fertilized central cell. After fertilization, auxin production triggers central cell proliferation and endosperm growth [36]. This coordination between fertilization and auxin production is accomplished by the specific repression of auxin biosynthetic genes, such as *YUC10* in maternal-derived tissues by the FERTILIZATION-INDEPENDENT SEED-PRC2 (FIS-PRC2) complex, which marks the target loci with H3K27me3 during the development of female gametes (Figure 1c, Table 1) [36,37]. As a consequence of heterochronic expression of *YUC10*, mutations affecting genes encoding subunits of the FIS-PRC2 complex, such as *FIS2*, *MEDEA* (*MEA*), *FERTILIZATION INDEPENDENT ENDOSPERM* (*FIE*), or *MULTICOPY SUPPRESSOR OR IRA1* (*MSI1*), result in fertilization-independent development of empty seeds. The auxin synthesized in the endosperm after fertilization is hypothesized to be transported to the outer ovule integument through auxin transporters, such as the ATP-binding cassette B10 (ABCB10), where it triggers seed coat differentiation by blocking PRC2 activity through a yet unknown mechanism [38,39]. Recent reports have also suggested the existence of redundant mechanisms driven by EMSY-like Tudor/Agenet H3K36me3 histone readers EMSY-Like protein 1 (EML1) and EML3, which actively repress auxin biosynthesis, as well as transport and signaling, during seed coat and endosperm development [40].

Rapid elongation of the hypocotyl is a typical shade-avoidance response in plants that requires auxin biosynthesis [41]. Recent reports have provided a link between the epigenetic regulation of auxin biosynthesis and shade-induced hypocotyl elongation. In light-grown seedlings, two AT-HOOK-CONTAINING NUCLEAR-LOCALIZED (AHL) proteins, AHL27 and AHL29, bind to the promoter of *YUC9* to repress its expression [42]. AHL29 has been shown to recruit ACTIN-RELATED PROTEIN 4 (ARP4), a member of the SWI2/SNF2-RELATED1 (SWR1) chromatin-remodeling complex that promotes nucleosome enrichment in the H2A variant H2A.Z, resulting in decreased accessibility of the RNA polymerase II and, therefore, gene repression (Figure 1d, Table 1) [42]. In response to shade, the PHYTOCHROME-INTERACTING FACTOR7 (PIF7) and H3K4me3/H3K36me3-binding protein Morf-Related Gene 2 (MRG2) bind to the *YUC8* promoter to facilitate the acetylation of H3 and H4, which promotes gene expression (Figure 1d, Table 1) [43].

Beyond covalent modifications of histones and nucleosome composition, another major epigenetic mechanism controlling the chromatin state, and hence gene expression dynamics, is the methylation of DNA cytosines. Three different cytosine methylation contexts are found in plants. Whereas methylation of cytosines is mostly restricted to CG dinucleotides in animals, plants can methylate cytosines in CG, CHG, and asymmetric CHH contexts (where H is A, T, or C), although CG methylation is the most frequent [44]. Two related DOMAINS REARRANGED METHYLTRANSFERASE (DRM1 and DRM2) participate in general de novo methylation, but different methyltransferases are involved in the maintenance of methylation depending on the cytosine context (reviewed in [11]). METHYLTRANSFERASE1 (MET1) maintains CG dinucleotide methylation, whereas CHROMOMETHYLASE3 (CMT3) is primarily involved in the maintenance of cytosine methylation in the CHG context; methylation in asymmetric CHH sequences is maintained by CMT2 and is related to stable silencing of transposable elements in heterochromatic regions [45]. DRM1 and DRM2, together with CMT3, participate in the methylation of both CHG and CHH sequences [46]. Loss-of-function alleles of *MET1* show a pleiotropic phenotype, including reduced fertility. Abnormal *met1* embryos have been shown to exhibit altered auxin distribution and expression of a major auxin transporter *PIN-FORMED1* (*PIN1),* but a direct relationship between CG methylation maintenance and *PIN1* gene expression control was not found [47]. Due to functional redundancy, mutations in *CMT3* and *DRM* genes do not lead to any visible phenotype. However, the triple mutant *drm1 drm2 cmt3* was shown to exhibit a pleiotropic phenotype with alterations during embryonic and postembryonic development similar to *met1* [46,48]. Most of the phenotypes found in this triple mutant, such as agravitropic root growth, leaf vascularity misconnection, and altered embryo development, were shown to be associated with alterations in auxin-related processes. Expression analyses have revealed that genes related to auxin biosynthesis (e.g., *YUC2* and *TAA1*) are upregulated in this mutant background. Interestingly, those transcriptional differences were mostly found in leaves, whereas almost no difference was found in roots, thus supporting a role for DNA methylation in tissue-specific patterns of gene expression. Additional experiments based on methylated DNA immunoprecipitation (Me-DIP)-PCR showed that DNA methylation levels at the *YUC2* promoter were reduced, suggesting direct regulation of auxin biosynthesis by non-CG DNA methylation in leaves (Figure 1e, Table 1) [48]. Further evidence of the *YUC2* regulation by DNA methylation was found in a search for temperature-dependent heterochromatin-associated short interfering RNAs (siRNAs) [49]. Plant asymmetric CHH methylation perpetuation through DNA replication requires 24 nt siRNAs to guide de novo cytosine methylation in a process called RNA-directed DNA methylation (RdDM) [11]. Gyula et al. reported a novel 24 nt siRNA named Locus_77297 in the regulatory region of *YUC2*. Levels of this small RNA and subsequent cytosine methylation were found to be decreased at 27 °C, which correlates with an increase in *YUC2* transcription [49].

Auxin homeostasis not only results from proper activation or repression of the biosynthetic routes. Most IAA molecules in cells are present as inactive forms [1]. Proper coordination between biosynthesis and inactivation allows cells to fine-tune the auxin concentration. IAA can be metabolically inactivated by (1) members of the UDP-glycosyltransferase (UGT) superfamily through the formation of ester-linked IAA–sugar conjugates, (2) members of the GRETCHEN HAGEN 3 (GH3) family via amide-linked IAA–amino-acid conjugates, (3) IAA carboxyl methyltransferases (IAMT) that produce methyl–IAA conjugates, and (4) oxidative inactivation by members of 2-oxoglutarate and iron-dependent dioxygenases, which convert IAA into 2-oxindole-3-acetic acid (oxIAA) [1,2]. In contrast to auxin biosynthesis, no epigenetic component has been reported to modulate auxin inactivation genes. Nevertheless, genome-wide studies have indicated that similar mechanisms may exist behind their regulation. For example, during leaf-derived callus formation, consistent inverted correlations between upregulation and decreased H3K27me3 were found for *GH3.1*, *GH3.2*, *GH3.3*, *GH3.6*, *GH3.17*, and *IAMT1* [16]. It was shown by Peng et al. that the *GH3.3* transcriptional response to shade was equivalent to *YUC8*, suggesting that analogous PIF7- and MRG2-driven mechanisms might operate to modulate *GH3.3* expression [43]. Further research is required to identify the precise epigenetic components regulating these and other genes for auxin metabolic inactivation.

## 3. Epigenetic Control of Auxin Transport

Cellular responses to auxin depend on its concentration. Besides the above-discussed role of the biosynthesis–inactivation balance, auxin transport plays a key role in auxin gradient formation [50,51,52]. The main protein families involved in auxin transport are (1) PIN-FORMED (PIN) proteins, which are typically polarly localized in the plasma membrane and act as auxin exporters; (2) PIN-LIKES (PILS) proteins, structurally related to PINs but located in the endoplasmic reticulum membrane; (3) AUXIN RESISTANT1 (AUX1) and LIKE AUX1 (LAX) influx carriers, which actively transport auxin from the apoplast into the cytoplasm; and (4) members of the type-B ATP-binding cassette family of proteins (ABCBs), which are located in the plasma membrane and can mediate auxin influx or efflux in a nonpolarized manner.

The first evidence of transcriptional control exerted by epigenetic mechanisms in auxin-transport-related genes, as auxin metabolic genes, came from genome-wide experiments. Characterization of epigenetic-driven cell reprogramming during pluripotency acquisition revealed a large decrease of H3K27me3 levels at *PIN1* in leaf-derived callus [16]. Similarly, comparative studies showed differential H3K27me3 levels in *PIN1*, *PIN4*, *PIN7*, and *PIN8* during leaf differentiation from the apical meristem [15]. Later, the *Arabidopsis* orthologue of the *Saccharomyces cerevisiae* SWI/SNF chromatin-remodeling ATPase BRAHMA (BRM) was found to directly target and positively regulate *PIN1*, *PIN2*, *PIN3*, *PIN4*, and *PIN7*, in part by antagonizing H3K27me3-associated chromatin repression mediated by PcG proteins (Figure 1g–i, Table 1) [53].

The most thoroughly characterized relationship between chromatin topology and auxin transport is regulation of the *PINOID* (*PID*) locus. PID is a serine/threonine kinase that phosphorylates PIN proteins to modulate their polar localization [54]. Expression of *PID* is regulated by the long intergenic noncoding RNA (lincRNA) AUXIN-REGULATED PROMOTER LOOP RNA (APOLO) in an auxin-dependent manner [55]. Knock-down of the *APOLO* loci results in a delayed gravitropic response resembling that of *pid* mutants. In the absence of an auxin stimulus, the upstream region of the *PID* locus forms a chromatin loop that blocks both *APOLO* and *PID* transcription. This loop is open upon auxin stimuli, allowing both *PID* and *APOLO* expression. Auxin-driven chromatin-loop-opening has been shown to be orchestrated by DNA demethylases DEMETER-LIKE 1 (DML1, also known as REPRESSOR OF SILENCING1 (ROS1)), DML2, and DML3 acting on the *APOLO* loci, although the mechanisms behind this regulation are not completely understood [55]. Concurrently, the *APOLO* lincRNA is transcribed and it physically associates to LHP1, triggering recruitment of the PRC2 machinery to re-establish the chromatin loop by increasing the repressive marks in the region (Figure 1f, Table 1) [55]. 

Recent reports have shed light on the interplay between specific H3K27me3 writers and erasers in regulating *PIN* expression during plant development. PRC2 has two redundant subunits with methyltransferase activity: CLF and SWINGER (SWN). Mutations in *SWN* do not generate any visible phenotype, but *clf* mutants show multiple and pleiotropic traits, ranging from leaf hyponasty to early flowering [56]. Interestingly, *clf* plants also exhibit a more branched root system with a higher density of lateral roots and longer primary root [57]. Auxin gradients and maxima are required for lateral root formation and PIN proteins are known to play an important role in this process [58]. Accordingly, it was found that CLF directly binds to the *PIN1* gene, repressing its expression and preventing lateral root formation (Figure 1g, Table 1) [57]. 

Several members of the Jumonji C (JmjC)-domain-containing-protein family regulate gene expression by direct histone demethylation antagonistically to PcG proteins. *Arabidopsis* EARLY FLOWERING 6 (ELF6/JMJ11), RELATIVE OF EARLY FLOWERING 6 (REF6/JMJ12), and JMJ30 have been shown to exhibit H3K27me3 demethylase activity [59,60,61]. In contrast to *CLF*, mutations in *REF6* cause simpler root architectures with fewer lateral roots [62]. Expression analyses have revealed that *PIN1, PIN2*, *PIN3*, *PIN4*, and *PIN7*, but not *AUX1*/*LAX* genes, are downregulated in a *ref6* background. REF6 was found to directly associate with *PIN1*, *PIN3*, and *PIN7* promoters through their CTCTGYTY motif, whereas the effect on *PIN2* and *PIN4* expression is most likely indirect (Figure 1g,h, Table 1) [62]. These findings are in line with previously described regulation of the *PIN* genes by BRM, which also antagonizes repressive methylation [53]. Indeed, it has been demonstrated that REF6 recruits BRM to cooperatively activate gene expression [63]. Remarkably, one of the genes with clear REF6-BRM coregulation is *YUC3*, an auxin biosynthetic gene without any previously reported epigenetic regulation [63].

Other chromatin remodelers and histone modification have been recently associated with the transcriptional status of *PIN1*. INOSITOL AUXOTROPHY 80 (INO80) is a member of the INO80 chromatin-remodeling factor family which regulates nucleosome structure and composition, and *NAP1-RELATED PROTEIN1* (*NRP1*) and *NRP2* code for two redundant histone chaperones [64,65]. A recent study has shown that triple mutants for these three genes exhibit auxin-deficiency-related phenotypes at inflorescences and roots. Tissue-specific chromatin immunoprecipitation analyses using tag-proteins revealed that INO80, NRP1, and NRP2 bind to *PIN1* regions to modulate its chromatin compactness (Figure 1g, Table 1) [66]. The subunit 2 of the largely evolutionarily conserved histone acetyltransferase complex ELONGATOR (ELP2) was shown to participate in different stress responses and developmental processes, including root meristem maintenance. Besides the regulation of important transcription factors for root apical meristem maintenance (e.g., *PLETHORA1* (*PLT1*), *PLT2*, *SCARECROW*, and *SHORTROOT*), *elp2* plants showed lower levels of auxin and *PIN1* downregulation at the root tip. This *PIN1* downregulation correlated with a decrease in H3 acetylation levels towards the 3′ coding region of the loci, suggesting epigenetic modulation of auxin transport and root growth by ELP2 [67]. 

## 4. Concluding Remarks, Open Questions, and Future Perspectives

The many roles of auxin in governing nearly every plant developmental program and integrating environmental inputs have attracted the attention of plant biologists for decades. This pivotal role of auxin relies on the correct activation/deactivation of its biosynthetic, transport, and signaling pathways through dynamic mechanisms, allowing plants to respond rapidly and accurately to their surroundings. Indirect observations anticipated that these responses might have an epigenetic nature [15,16]. In the past five years, important traits of high agronomic relevance, for example, endosperm and seed coat development; floral whorl patterning; root branching; and responses to gravity, shade, and temperature, have been found to be linked to proper epigenetic control of specific genes for auxin biosynthesis and transport in *Arabidopsis* [35,36,38,42,43,48,49,55,57,62]. Members of the five main groups of epigenetic-related proteins (chromatin remodelers, histone modifiers, Polycomb or Polycomb-related proteins, enzymes involved in DNA methylation, and members of RdDM) were found to participate in the epigenetic control of auxin homeostasis during a wide range of developmental processes (Figure 1, Table 1).

Nevertheless, there are still substantial gaps in knowledge regarding how auxin and the epigenetic machinery interact with each other. Several lines of evidence suggest that auxin is able to abolish PRC2 activity at the transcriptional level during seed coat development, but the precise mechanisms are unexplored [38]. This striking observation might constitute the first clue about how PRC2 genes, which are developmental transition modulators, are regulated, and it may represent a common mechanism for other developmental stages.

Functional redundancy among gene families, such as YUC or PIN, challenges the genetic dissection of individual gene roles. The recent findings reviewed here have helped to elucidate the roles of specific members of the YUC and PIN families in plant development, ranging from a very specific role for *YUC10* in endosperm development [36] to a more general role for *YUC2* in leaf growth [48]. At the same time, genome-wide occupancy data suggest that the *YUC3* gene is intimately regulated by the histone demethylase REF6 and the chromatin-remodeler BRM, although the process for which this regulation is relevant remains to be determined [63].

Out of the large number of genes known to participate in auxin homeostasis, there are only a few for which direct epigenetic regulation has been described, revealing the widespread poor understanding of this topic. The most obvious lack of data appears to be within the auxin inactivation gene families. While some of these genes, such as *DIOXYGENASE FOR AUXIN OXIDATION 1* (*DAO1*) and *DAO2,* have been recently identified [18,68,69], others, such as *GH3* genes, were identified as auxin-responsive genes more than 30 years ago. Further research aimed at deciphering the epigenetic modulation of these genes might open up possibilities for modifying auxin homeostasis and many agronomical traits via auxin inactivation pathways.

One of the best characterized interplays between histone-modifying enzymes was found to affect a very important trait for plant fitness, namely, lateral root development. The density and length of lateral roots directly impact a plant’s ability for nutrient and water uptake. CLF and REF6 antagonistically affect lateral root development by depositing or removing the repressive mark H3K27me3, mainly on the *PIN1* gene [57,62]. Mutations in other auxin transporters, such as *AUX1* or *LAX3*, are known to alter lateral root patterning, but a direct role of CLF or REF6 in their regulation has not yet been explored. The possibility that these proteins exert a regulatory role in other auxin homeostatic genes cannot be ruled out. The recently described negative correlation between DNA methylation and REF6 binding capacity adds an additional layer of complexity, interconnecting cytosine methylation with histone modifications, which is a fascinating subject requiring further study in plants [70].

Eukaryotic DNA methylation is typically associated with gene silencing, with an important role in transposon inactivation. This mechanism is particularly relevant in plants in which cytosines in all possible contexts can be methylated. Different enzymes have been found to catalyze methylation depending on the genomic situation. Altered expression levels/patterns of specific *PINs* have been shown in *met1* embryos, as well as in the leaves and roots of the *drm1 drm2 ctm3* triple mutant [47,48]. Although a direct mechanism connecting DNA methylation and modulation of auxin homeostatic genes has not yet been found, recent findings linking *YUC2* expression to RdDM [49] encourages further research towards this line of evidence. Moreover, a DNA demethylation-dependent mechanism involving the lincRNA *APOLO* has been shown to regulate the *PID* loci, which in turn modulates PIN function [55]. This apparently warped mechanism has raised new intriguing questions regarding the exact mechanisms and specific components driving this auxin-dependent DNA demethylation.

Despite the considerable amount of literature regarding auxin biosynthesis, transport, and signaling, very little is known about the control of the expression dynamics of the genes involved in these processes. However, the few reports available have indicated that this complex and interconnected hormonal pathway is subject to epigenetic cues, providing additional elements that favor developmental plasticity. Further multidisciplinary efforts are imperative to better understand the different modulators of hormone action. This knowledge will be instrumental for the eventual manipulation of hormone homeostasis in order to improve plant fitness and adaptation.

## Figures and Tables

**Figure 1 biomolecules-09-00623-f001:**
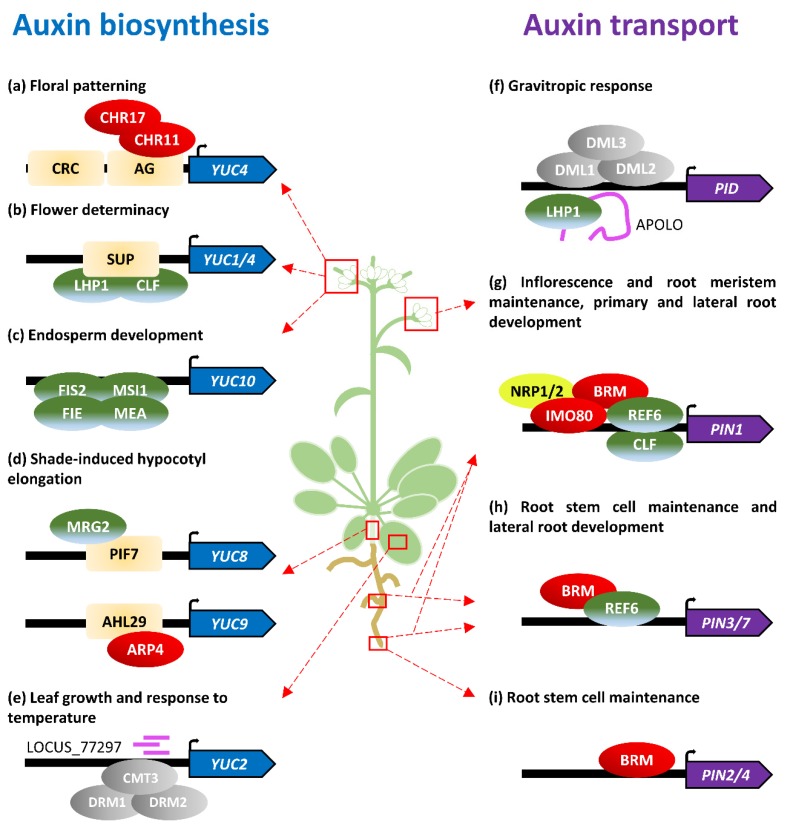
Auxin biosynthesis and transport are epigenetically regulated along the plant life cycle. (**a**–**h**) Epigenetic regulation of (**a**–**e**) auxin-biosynthesis-related genes and (**f**–**i**) auxin-transport-related genes. Colored ovals represent proteins of the epigenetic machinery: chromatin remodelers (red), Polycomb and other histone modifiers (green), DNA methylation/demethylation enzymes (grey), histone chaperones (yellow), and noncoding RNAs (pink). Pale orange rectangles indicate transcription factors. Proteins promoting gene activation are represented above the gene, whereas factors drawn below represent repressors. Distances to the regulated gene are not drawn to scale.

**Table 1 biomolecules-09-00623-t001:** Auxin homeostasis-related genes with directly proven epigenetic regulation.

	Gene Name	AGI Code	Regulated by	Developmental or Response to Ambient Process	Reference
Auxin biosynthesis	*YUC1*	At4g32540	CLF, LHP1	Floral patterning	[35]
	*YUC2*	At4g13260	DRM1, DRM2, CMT3, Locus_77297	Leaf growth and high-temperature response	[48,49]
	*YUC3*	At1g04610	BRM, REF6	n.s.	[63]
	*YUC4*	At5g11320	CLF, LHP1, CHR11, CHR17	Floral patterning and floral determinacy	[31,35]
	*YUC8*	At4g28720	MRG2	Shade-induced hypocotyl elongation	[43]
	*YUC9*	At1g04180	ARP4	Shade-induced hypocotyl elongation	[42]
	*YUC10*	At1g48910	FIS2-PRC2	Endosperm development	[36]
Auxin transport	*PID*	At2g34650	DML1, DML2, DML3, LHP1, lincRNA-APOLO	Gravitropic response	[55]
	*PIN1*	At1g73590	BRM, CLF, REF6, INO80, NRP1, NRP2, ELP2	Inflorescence and root meristem maintenance, primary and lateral root development	[53,57,62,66,67]
	*PIN2*	At5g57090	BRM	Root stem cell maintenance	[53]
	*PIN3*	At1g70940	BRM, REF6	Root stem cell maintenance, lateral root formation	[53,62]
	*PIN4*	At2g01420	BRM	Root stem cell maintenance	[53]
	*PIN7*	At1g23080	BRM, REF6	Root stem cell maintenance, lateral root formation	[53,62]

ARP4, ACTIN-RELATED PROTEIN 4; BRM, BRAHMA; CLF, CURLY LEAF; CHR11, CHROMATIN REMODELING 11; CMT3, CHROMOMETHYLASE3; DML1, DEMETER-LIKE 1; DRM1, DOMAINS REARRANGED1; ELP2, ELONGATOR COMPLEX SUBUNIT2; FIS2, FERTILIZATION INDEPENDENT SEED2; INO80, INOSITOL AUXOTROPHY80; LHP1, LIKE HETEROCHROMATIN 1; lincRNA-APOLO; long intergenic noncoding RNA-APOLO; MRG2, MORF-RELATED GENE 2 (MRG2); NRP1, NAP-RELATED PROTEIN1; PRC2, POLYCOMB REPRESSIVE COMPLEX 2; REF6; RELATIVE EARLY FLOWERING 6; n.s. stands for nonstudied.

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
