# Peer review of "Epigenetic Regulation of Auxin Homeostasis"

_biomolecules, 2019, doi:10.3390/biom9100623_

Round 1

Reviewer 1 Report

It is a well written, interesting, and thorough review of the epigenetic regulation of auxin homeostasis-regulating genes, therefore, I support its publication in Biomolecules after correcting a few typos.

I found only some minor errors:

Line 168, 174, 179 and Table 1: DMR is misspelt, correctly it is DRM.

Line 192: possibly there are some extra spaces in this line. Please check.

Author Response

Reviewer #1:

-It is a well written, interesting, and thorough review of the epigenetic regulation of auxin homeostasis-regulating genes, therefore, I support its publication in Biomolecules after correcting a few typos.

Thank you very much for your positive comments on our manuscript.

I found only some minor errors:

-Line 168, 174, 179 and Table 1: DMR is misspelt, correctly it is DRM.

We have corrected all these typos.

-Line 192: possibly there are some extra spaces in this line. Please check.

We have corrected this mistake.

Reviewer 2 Report

The review article by Mateo-Bonmati et al is very well-written and informative. The manuscript highlights an exciting and complex growing field that will likely prove critical in developing a full understanding of auxin biology. I only have a couple of minor comments/suggestions:

Lines 35-41: The recently published work by Powers et al. (2019 Mol Cell) suggests there may be a more complicated mechanism that what is described here - whereby high local auxin concentrations result in the translocation of ARFs from the cytoplasm to the nucleus. It would be informative to allude to this here or in the conclusions to highlight the complex regulation of auxin signaling (although I understand this is not the focus of the review). Lines 51 and 56: Histone Acetyltransferases is capitalized in 51 and not 56. Probably both should be lower-case. There is no mention of indole-3-butyric acid in the manuscript. It would be appropriate to at least mention IBA in the discussion of epigenetic regulation.

Author Response

Reviewer #2:

-The review article by Mateo-Bonmati et al is very well-written and informative. The manuscript highlights an exciting and complex growing field that will likely prove critical in developing a full understanding of auxin biology. I only have a couple of minor comments/suggestions:

Thank you very much for your positive comments on our manuscript. 

-Lines 35-41: The recently published work by Powers et al. (2019 Mol Cell) suggests there may be a more complicated mechanism that what is described here - whereby high local auxin concentrations result in the translocation of ARFs from the cytoplasm to the nucleus. It would be informative to allude to this here or in the conclusions to highlight the complex regulation of auxin signaling (although I understand this is not the focus of the review).

We agree with reviewer #2 that we succinctly introduce the auxin signalling, but that we did it on purpose. Besides the suggested excellent piece of work on posttranslational modifications and subcellular localization of ARFs (Powers et al. 2019), we have also omitted many nice papers on how ARFs are posttranscriptionally regulated by small RNAs, such as microRNAs or tasiRNAs. We aimed to keep the focus on known epigenetic mechanisms regulating genes involved in IAA metabolism and transport, and not to expand in excess the introduction with very interesting stories that might be out of the scope of this review.

-Lines 51 and 56: Histone Acetyltransferases is capitalized in 51 and not 56. Probably both should be lower-case. 

We have corrected this mistake. Now “histone acetyltransferase” is written in lower-case along the manuscript.

-There is no mention of indole-3-butyric acid in the manuscript. It would be appropriate to at least mention IBA in the discussion of epigenetic regulation.

We acknowledge this appreciation. IBA has been involved in many plant developmental responses, likely due to a rapid IBA-to-IAA conversion. However, in order to keep the focus on the main topic of the review, we consider expendable to mention other auxin forms such as IBA, PAA or 4-Cl-IAA.